# Efficacy of oxfendazole treatment against infective larvae of *Litomosoides sigmodontis*

Frederic Risch[1,2], Lucas J. Bergmann[1¤], Hannah R. Wegner[1], Achim Hoerauf[1,2], Marc P. Hübner 🔘[1,2]*

**1** University of Bonn, University Hospital Bonn, Institute for Medical Microbiology, Immunology and Parasitology, Bonn, Germany, **2** German Center for Infection Research, Partner site Bonn-Cologne, Bonn, Germany

¤ Current address: Department of Internal Medicine, University Medical Center Hamburg-Eppendorf, Hamburg, Germany
* huebner@uni-bonn.de

## Abstract

Filarial infections are prevalent worldwide and affect both humans and animals. Two filarial infections, lymphatic filariasis and onchocerciasis, are listed among the neglected tropical diseases by the WHO and targeted for their elimination via large-scale public health campaigns in the affected countries. New drugs are urgently needed to facilitate this goal, and the WHO has outlined the need to develop novel adult worm killing (=macrofilaricidal) compounds for onchocerciasis as one of the critical actions in their roadmap for neglected tropical diseases for 2020–2030. Oxfendazole (Oxf) is an anthelmintic benzimidazole that has been used in the veterinary field for decades and is currently being repurposed for human filarial infections. Previous work has shown that orally administered Oxf has a strong activity against the adult worms of *Litomosoides sigmodontis*. Based on these and other findings, Oxf was selected for further clinical development, and a phase I clinical trial conducted in 2019 reported no significant safety issues. Following these results, a multi-centric phase II basket trial was designed and is conducted by the eWHORM consortium (https://ewhorm.org/). In this study we investigated the efficacy of Oxf against the infective larval stage of *L. sigmodontis* in BALB/cJ mice. Female BALB/cJ mice were naturally infected and treated orally with Oxf starting 1 day after infection for up to 5 days. Mice were euthanized 35 days after infection and the number and length of worms were assessed *ex vivo*. Oxf treatment for 5 days led to near-complete clearance of worms (97.6% median reduction compared to vehicle control) while a 3-day treatment regimen demonstrated a dose-dependent reduction in recovered worms (up to 82.6%). The length of isolated worms was reduced to a similar degree irrespective of treatment duration. These results indicate that treatment with Oxf can clear early filarial infections in addition to its known macrofilaricidal activity.

**Data availability statement:** All original data are provided as excel sheet (Supporting information).

**Funding:** HRW was supported by the Jürgen Manchot Stiftung. AH and MPH received funding via Germany's Excellence Strategy (EXC2151-390873048), MPH received funding via the German Center for Infection Research (TTU 09.701, 09.733). FR and LJB did not receive funding for this study. The funders had no role in study design, data collection and analysis, decision to publish, or preparation of the manuscript.

**Competing interests:** The authors have declared that no competing interests exist.

## Author summary

Filarial worms can cause debilitating diseases and affect millions of people worldwide. New drugs are urgently needed to facilitate the elimination of filarial infections and improve the well-being and health of affected patients. Oxfendazole is one such drug that has been used in the veterinary field for more than 40 years and is currently being repurposed for human use. Previous studies have shown that oxfendazole can eliminate adult worms in a filarial rodent model. To support the further development of oxfendazole, we investigated its efficacy against the infective larval stage using the rodent filarial nematode *Litomosoides sigmodontis*. We demonstrate that early treatment with oxfendazole for three or five days leads to a strong reduction of isolated worms by >80% and 95%, respectively. These results indicate that treatment with Oxf clear early filarial infections in addition to its known activity against adult worms.

## Introduction

Filarial worms are prevalent worldwide and can cause severe diseases in humans and animals. The most significant infections in humans are lymphatic filariasis (caused by *Wuchereria bancrofti*, *Brugia malayi* and *Brugia timori*), onchocerciasis (*Onchocerca volvulus*), loiasis (*Loa loa*) and mansonellosis (*Mansonella* spp.) [1]. Out of those four, lymphatic filariasis and onchocerciasis are listed among the neglected tropical diseases by the WHO. Lymphatic filariasis and onchocerciasis are targeted for elimination as a public health problem or elimination of transmission, respectively, via large-scale public health campaigns in the affected countries [2]. New drugs are urgently needed to facilitate this goal and the WHO has outlined the need to develop novel compounds for onchocerciasis as one of the critical actions in their neglected tropical diseases roadmap for 2021–2030 [2].

Oxfendazole (Oxf) is an anthelmintic compound that has been used in the veterinary field for more than 40 years [3]. Oxf belongs to the class of benzimidazole compounds and various groups have been working on the repurposing of Oxf for human filarial infections [4–6]. Previous work has shown that orally administered Oxf has a strong activity against the adult worms (=macrofilaricidal activity) of *L. sigmodontis* [4]. In addition, *in vitro* data suggested activity against adult life cycle stages of *Onchocerca* spp. [4]. Based on these and other indications, Oxf was selected for further clinical development and a phase I clinical trial was conducted in 2019 in which no significant safety issues were noted [6]. Following these results, a multi-centric phase II basket trial was designed that includes participants with onchocerciasis, loiasis, mansonellosis as well as trichuriasis and is currently conducted by the eWHORM consortium (https://ewhorm.org/). In addition to its development for human patients, results from a study in one of the recently published novel mouse models for the canine heartworm demonstrate the efficacy of Oxf against the developing stages of *Dirofilaria immitis* [7].

Previous research has shown that Oxf has a strong macrofilaricidal activity in the *L. sigmodontis* rodent model and no direct activity against the first larval stage (=microfilariae) of *L. sigmodontis* [4] or *L. loa* [8]. However, the efficacy against the infective larval stage (L3) that is transmitted by the vector has not been previously studied. In this short communication, we report that Oxf has a strong efficacy against the L3 larval stage and may thus prevent the development of chronic infections after exposure to the vector.

## Materials and methods

### Ethics statement

All animal experiments were performed according to EU Directive 2010/63/EU and approved by the appropriate state authority "Landesamt für Natur-, Umwelt- und Verbraucherschutz" (now known as "Landesamt für Verbraucherschutz und Ernährung"), Recklinghausen, Germany (Permit: 81-02.04.2020.A244).

### Animals and natural infection with *Litomosoides sigmodontis*

6-week old female BALB/cJ wild-tpye (WT) mice were purchased from Janvier Labs, Saint-Berthevin, France. For the experiments, mice were housed in individually ventilated cages with unlimited access to food and water and a 12-hour day/night cycle within the animal facility at the Institute for Medical Microbiology, Immunology and Parasitology (IMMIP), University Hospital Bonn.

6-8 week old female mice were naturally infected with *L. sigmodontis* via exposure to the tropical rat mite, *Ornithonyssus bacoti*, carrying the infective L3 larvae. In short, mice were placed in cages with bedding material containing the mites. After 24 hours, the bedding material containing the mites was removed, and the cages were placed on top of a plastic tub with disinfectant below the cages and no direct contact with the mice. After an additional 24 hours, mice were moved into standard cages, and the bedding material was exchanged daily for five days to remove any remaining mites.

### Treatment

For the Oxf treatment, a commercially available formulation of Oxf (Dolthene) was used and dissolved in corn oil (Sigma-Aldrich, St. Louis, USA). Vehicle controls received only corn oil. For all experiments, treatment with Oxf was performed orally, twice per day, 8 hours apart starting 1 day post infection (dpi). Mice received either 5 or 12.5 mg/kg Oxf in a total volume of 5 ml/kg per treatment for 3 or 5 days as indicated in Figs 1 and 2. Daily doses were selected based on previously demonstrated efficacy against adult worms of *L. sigmodontis* [4,9,10]. The treatment was well tolerated by the animals and no adverse events were noted.

### Parasite recovery and quantification

Necropsies were performed to quantify the worm burden 35 dpi to coincide with the development of adult worms in this model [11,12]. Mice were euthanized with an overdose of isoflurane, and worms were isolated via lavage of the thoracic cavity with 8–10 ml PBS (Thermo Fisher Scientific, Waltham, USA). Isolated worms were counted, measured (up to 10 worms per gender per mouse) and the gender was determined.

### Statistical analysis

Statistical analysis was performed with GraphPad Prism software version 10 (GraphPad Software, San Diego, USA). Data was assumed to follow a non-parametric distribution due to the low sample size. Kruskal-Wallis test followed by Dunn's multiple comparisons test was used to assess significant differences between 3 or more groups (in case of >2 groups, groups were compared to the vehicle control only) or the Mann-Whitney-U test for differences between two groups. Data

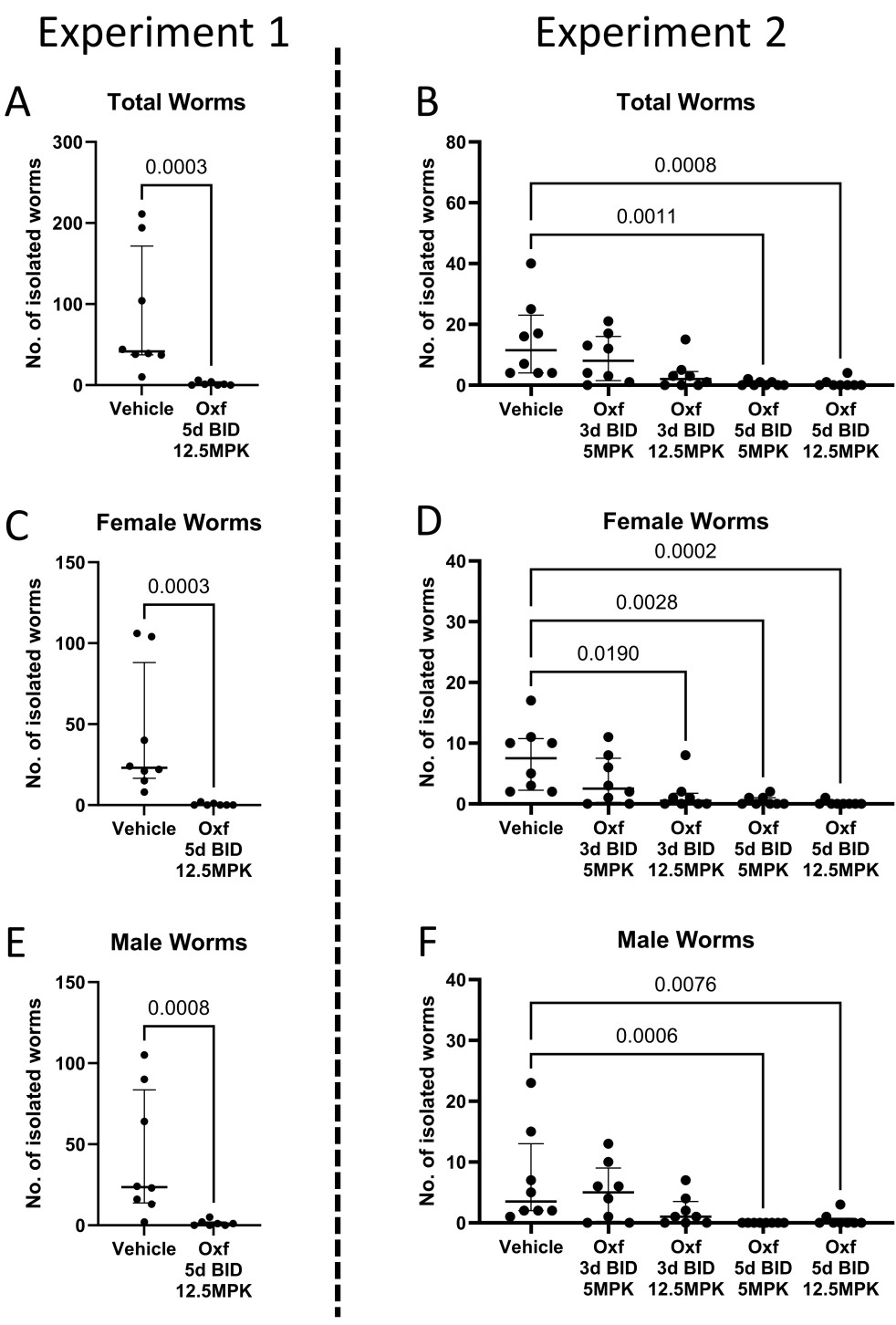

**Fig 1. Number of isolated worms after treatment with oxfendazole.** 6-8 week old female BALB/cJ mice were naturally infected with *Litomosoides sigmodontis* and then treated orally with oxfendazole (Oxf) twice daily starting 1 day after the infection (dpi). In the first experiment (A, C, E), mice were either treated with 12.5 mg/kg Oxf for 5 days or the vehicle control for the same amount of time. In the second experiment (B, D, F), mice were treated with either 5 or 12.5 mg/kg Oxf for 3 or 5 days or the respective vehicle control. Mice were euthanized 35 dpi and worms were isolated from the pleural cavity before quantification. Data is shown as number of isolated worms per mouse with lines indicating median and interquartile range. Statistical analysis was performed with either Mann-Whitney U test (experiment 1) or Kruskal-Wallis test followed by Dunn's post test (experiment 2). (A+B) Total number of isolated worms; (C+D) number of isolated female worms; (E+F) number of isolated male worms. MPK=mg/kg bodyweight.

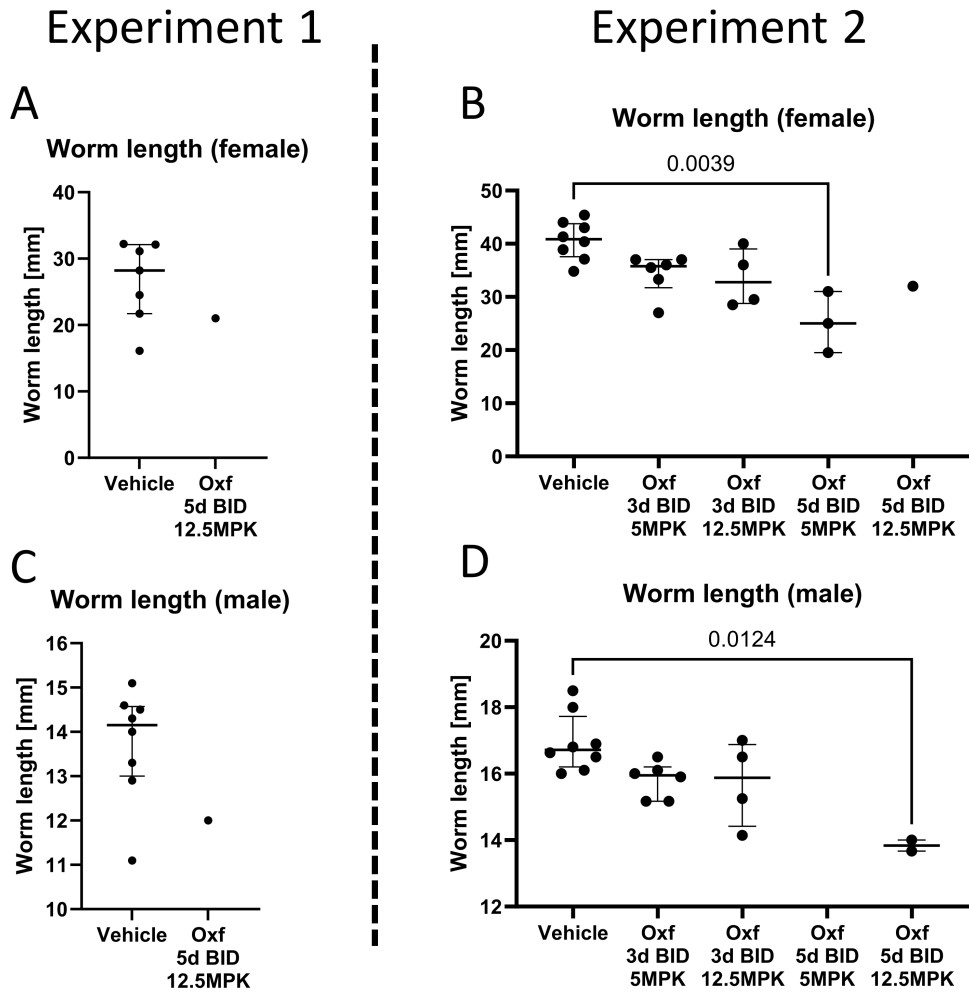

**Fig 2. Worm length of isolated worms after treatment with oxfendazole.** 6-8 week old female BALB/cJ mice were naturally infected with *Litomosoides sigmodontis* and then treated orally with oxfendaole (Oxf) twice daily starting 1 day after the infection (dpi). In the first experiment (A+C), mice were either treated with 12.5 mg/kg Oxf for 5 days or the vehicle control for the same amount of time. In the second experiment (B+D), mice were treated with either 5 or 12.5 mg/kg Oxf for 3 or 5 days or the respective vehicle control. Mice were euthanized 35 dpi and worms were isolated from the pleural cavity and then measured. Data is shown as mean worm length per mouse with lines indicating median and interquartile range. Statistical analysis was performed with either Mann-Whitney U test (experiment 1) or Kruskal-Wallis test followed by Dunn's post test (experiment 2). (A+B) Length of female worms; (C+D) length of male worms. MPK = mg/kg.

are shown as median with interquartile range. Only p values < 0.05 are shown in the graphs. The original data sets are shown in S1 Data.

## Results

Female BALB/cJ mice were naturally infected with *Litomosoides sigmodontis* in two independent experiments. Mice were treated with Oxf or a vehicle control one day after the infection (dpi) to assess the efficacy of Oxf against the infective L3 larval stage. Treatment was administered orally twice daily for either 3 or 5 days. Necropsies were performed 35 dpi to assess the development of remaining worms into later stages as well as quantify (Fig 1) and measure (Fig 2) all remaining worms.

A five-day treatment with 2x12.5 mg/kg Oxf led to a statistically significant median reduction of isolated worms by 97.6% compared to the vehicle control (Fig 1A). Both female and male worms were reduced to a similar degree, with median reductions of 100% and 95.7%, respectively (Fig 1C and 1E).

Following these results, a second experiment was performed to assess the efficacy of a lower daily dosage (2x 5 mg/kg) as well as a shorter treatment regimen (3 rather than 5 days, Fig 1B/1D/1F). The efficacy of the 5-day treatment regimen was confirmed in the second experiment. In addition, a dose-dependent reduction of the number of isolated worms was seen for the 3-day treatment. While the 2x5 mg/kg treatment led to only a slight reduction of isolated worms (30.4% compared to the vehicle control), the 2x12.5 mg/kg treatment reduced the number of total worms by 82.6% ($p = 0.059$), female worms by 93.3% ($p = 0.019$) and male worms by 71.4% ($p = 0.258$).

Next, we measured the isolated worms to determine an impact of the treatment on their development (Fig 2). Despite the low number of isolated worms in some of the treatment groups, it appears that the 5-day treatment regimen induced a reduction of worm length ($p < 0.05$ with 2x5 mg/kg Oxf) that seems to be less pronounced in the 3-day treatment groups (Fig 2B and 2D).

Overall, oral administration of Oxf was shown to be highly effective against the infective L3 stage of *L. sigmodontis*.

## Discussion

Oxf is a repurposed benzimidazole compound that is currently in clinical development for filarial infections as well as trichuriasis and other soil-transmitted helminths in humans [13]. In addition, recent studies have suggested that Oxf could also be employed to target the canine heartworm, *D. immitis* [7]. Previous work by our group has shown that Oxf has a stage-specific activity in the *L. sigmodontis* mouse model [4]. Oral administration of Oxf was shown to be highly effective against the adult worms while at the same time it was demonstrated that Oxf exhibits no direct activity against the first larval stage (microfilariae). This lack of microfilaricidal activity is a significant advantage of Oxf compared to some other compounds since the rapid death of microfilariae is associated with the release of large amounts of filarial and, dependent on the filarial species, bacterial antigen (via their *Wolbachia* endosymbionts) that in turn induces strong inflammatory reactions [14–17].

However, the efficacy against the infective larval stage (L3) that is transmitted by the vector has not been previously studied. In this study, we report that Oxf exhibits a strong efficacy against the L3 larval stage and may thus prevent the development of chronic infections after exposure to the vector. While the half-life of orally administered Oxf is fairly short (~1.8 – 2.8 h in mice and 9.2 – 11.8 h in Caucasians or 11.6 – 13.9 h in healthy Africans) and Oxf is thus unlikely to prevent re-infections for a significant amount of time [4,6,18], the herein described results suggest that Oxf treatment might clear an ongoing, recently contracted filarial infection.

In addition, the five-day Oxf treatment regimen with either 5 mg/kg or 12.5 mg/kg BID is sufficient to clear not only the L3 larvae but also the adult of *L. sigmodontis* as shown in [4]. In contrast, while the tested three-day Oxf regimen was enough to clear the L3 larvae in this study, a similar treatment regimen (12.5 mg/kg BID for three days) had no statistically significant impact on the number of recovered adult worms compared to a vehicle control as previously published [9], indicating a higher susceptibility of the third larval stage. The susceptibility of the *L. sigmodontis* fourth larval stage to Oxf remains to be analyzed experimentally to confirm that both recently acquired and chronic filarial infections are cleared by Oxf. However, data from *D. immitis*-infected NSG mice demonstrate that Oxf (5 mg/kg BID) administered against the third (1 dpi) and fourth larval stages (29 dpi) significantly reduced the L4 larval burden [7]. Based on these findings, we hypothesize that a three- to five-day Oxf treatment would also be effective against the fourth larval stage of *L. sigmodontis*.

In addition, data from the first phase 1 clinical trial suggests that the PK properties of Oxf are quite atypical for benzimidazole anthelmintics and enable much higher systemic exposure (including higher $C_{max}$ and half-life) than for example albendazole which is currently being used for the treatment of lymphatic filariasis [6]. Thus, it is likely that comparable dosages would lead to a higher efficacy than the current standard of care.

Assmus et al. recently published PK results of a study in healthy African volunteers for an Oxf tablet formulation that was developed for the treatment of onchocerciasis [19]. Based on these PK results, the authors simulated various dosages and the likelihood that these regimens would achieve a plasma concentration >200 ng/mL for a minimum of 5 days, which was postulated to be a sufficient dose to clear the adult worms of *Onchocerca volvulus*. Using standard allometric scaling, a 5 mg/kg and 12.5 mg/kg dose in mice is equivalent to a human dose of 0.41 and 1.01 mg/kg. Based on the modelling by Assmus et al., the herein tested 5 mg/kg BID (equivalent to about 25 mg BID for a 60 kg human) regimen would result in a 80% chance of reaching the required plasma levels while the 12.5 mg/kg BID (equivalent to ca. 61 mg BID for a 60 kg human) would have a chance of >92.6% [4,9,19]. By comparison, the dosages selected for the ongoing phase II trial for onchocerciasis are 400 mg and 800 mg QD administered for 5 days (PACTR202412611774752). While these dosages are significantly higher, modelling by Assmus et al. suggests that these treatment regimens have a comparable chance of reaching the targeted plasma level of Oxf as the above mentioned 61 mg BID dose (91.7% for the 400 mg QD group and 95.9% for the 800 mg QD group) [19].

Next to the development for human filarial infections, Oxf may also be a potential drug candidate to treat migrating *D. immitis* larvae and prevent canine heartworm disease. Given the risk of life-threatening adverse events following the death of *D. immitis* adult worms, such a treatment requires careful consideration and implementation. As such, the target product profile for any new drug is generally strict for *D. immitis* and aims for a 100% prevention of adult worm development. Further dose-finding studies based on the novel mouse models (summarized by Dagley et al. in [7]) are needed here to ascertain the use of Oxf to treat canine heartworms.

## Supporting information

**S1 Data. Original data sets shown in Figs 1 and 2.**
(XLSX)

## Acknowledgments

The authors would like to thank Benjamin Lenz, Marianne Koschel and Tilman Aden for technical assistance with the mouse experiments.

## Author contributions

**Conceptualization:** Frederic Risch, Marc P. Hübner.

**Data curation:** Frederic Risch.

**Formal analysis:** Frederic Risch, Lucas J. Bergmann.

**Funding acquisition:** Marc P. Hübner.

**Investigation:** Frederic Risch, Lucas J. Bergmann, Hannah R. Wegner.

**Methodology:** Frederic Risch.

**Project administration:** Marc P. Hübner.

**Resources:** Achim Hoerauf, Marc P. Hübner.

**Supervision:** Marc P. Hübner.

**Validation:** Frederic Risch.

**Visualization:** Frederic Risch, Lucas J. Bergmann.

**Writing – original draft:** Frederic Risch.

**Writing – review & editing:** Frederic Risch, Lucas J. Bergmann, Hannah R. Wegner, Achim Hoerauf, Marc P. Hübner.

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
