## [Decision Letter · Decision Letter 0]

16 Feb 2026

Response to Reviewers'. This file does not need to include responses to any formatting updates and technical items listed in the 'Journal Requirements' section below.'. This file does not need to include responses to any formatting updates and technical items listed in the 'Journal Requirements' section below.* A marked-up copy of your manuscript that highlights changes made to the original version. You should upload this as a separate file labeled 'Revised Manuscript with Track Changes'.'.* An unmarked version of your revised paper without tracked changes. You should upload this as a separate file labeled 'Manuscript'.'.If you would like to make changes to your financial disclosure, competing interests statement, or data availability statement, please make these updates within the submission form at the time of resubmission. Guidelines for resubmitting your figure files are available below the reviewer comments at the end of this letter.We look forward to receiving your revised manuscript.Kind regards,Bruce A. RosaAcademic EditorPLOS Neglected Tropical DiseasesLucienne TrittenSection EditorPLOS Neglected Tropical Diseases

Shaden Kamhawi

co-Editor-in-Chief

Paul Brindley

co-Editor-in-Chief

**Additional Editor Comments:**Overall, the reviewers were positive about the manuscript. They have provided a few suggested revisions, so please carefully consider their feedback.**Journal Requirements:**

3) We note that your Data Availability Statement is currently as follows: "All original data will be provided as excel sheet.". Please confirm at this time whether or not your submission contains all raw data required to replicate the results of your study. Authors must share the “minimal data set” for their submission. PLOS defines the minimal data set to consist of the data required to replicate all study findings reported in the article, as well as related metadata and methods (https://journals.plos.org/plosone/s/data-availability#loc-minimal-data-set-definition).

- The points extracted from images for analysis..

If your submission does not contain these data, please either upload them as Supporting Information files or deposit them to a stable, public repository and provide us with the relevant URLs, DOIs, or accession numbers. For a list of recommended repositories, please see https://journals.plos.org/plosone/s/recommended-repositories

4) Please amend your detailed Financial Disclosure statement. This is published with the article. It must therefore be completed in full sentences and contain the exact wording you wish to be published.

**Reviewers' comments:**Reviewer's Responses to Questions

**Key Review Criteria Required for Acceptance?**

**Methods**

-Are the objectives of the study clearly articulated with a clear testable hypothesis stated?

-Is the study design appropriate to address the stated objectives?

-Is the population clearly described and appropriate for the hypothesis being tested?

-Is the sample size sufficient to ensure adequate power to address the hypothesis being tested?

-Were correct statistical analysis used to support conclusions?

-Are there concerns about ethical or regulatory requirements being met?

Reviewer #1: The Methods are well-described and appropriate for the aims of the study.

Reviewer #2: The paper describes that Oxfendazole exhibits a robust efficacy against the L3 larval stage of Litomosoides sigmodontis in mice. This work supports the ongoing clinical development of Oxfendazole as a broad stage filaricide/nematicide. The benzimidazole originates from veterinary medicine and is now repurposed for human use. While the team showed in previous work, that Oxfendazole showed macrofilaricidal efficacy, but no direct effect on microfilariae, only this manuscripts demonstrates that Oxfendazole is active against L3 larvae.

The experimental design is sufficiently described and scientifically sound:

• BALB/cJ mice were naturally infected with L. sigmodontis via exposure to infected mites.

• Drug was administered orally twice daily, starting 1 day post infection, for 3 or 5 days.

• Doses tested: 5 mg/kg and 12.5 mg/kg.

• Worms were recovered and measured 35 days post infection.

**Results**

-Does the analysis presented match the analysis plan?

-Are the results clearly and completely presented?

-Are the figures (Tables, Images) of sufficient quality for clarity?

Reviewer #1: The Results are clear and the analyses justified and appropriate.

Reviewer #2: Key Findings

1. Strong reduction in worm burden

• 5 day treatment at 12.5 mg/kg: → 97.6% median reduction in total worms → Female worms reduced 100%, males 95.7%

• 3 day treatment showed dose dependent efficacy: → 12.5 mg/kg for 3 days reduced total worms by 82.6% → 5 mg/kg for 3 days had only modest effect (~30% reduction)

2. Worm development was impaired

• Surviving worms were significantly shorter, especially after 5 day treatment.

• Oxfendazole is highly effective against the infective L3 stage of L. sigmodontis.

**Conclusions**

-Are the conclusions supported by the data presented?

-Are the limitations of analysis clearly described?

-Do the authors discuss how these data can be helpful to advance our understanding of the topic under study?

-Is public health relevance addressed?

Reviewer #1: The regimen investigated here was clearly highly effective against L3 stage parasites. My concern is the seemingly minor relevance. To be useful, one would either have to maintain constant plasma levels in people living in endemic regions or treat individuals who were briefly exposed to transmission. Activity against L4 and L5 stages might be more clinically relevant (not considering macrofilaricidal activity, which has been reported).

Reviewer #2: I would encourage the authors to extend the background and discussion about animal health as well. To prevent canine heartworm disease, the migrating larvae need to be treated. This could also lead to an outlook on what the treatment against L4 larvae would look like. Furthermore, the authors may want to discuss the dose-response comparing the larvicidal versus the adulticidal effects to a certain degree (particularly as parasite isolate, mouse strain and drug are uniform).

Minor remark: The graphs and abstract do not clearly state that the treatment is 2 times per day.

**Editorial and Data Presentation Modifications?**

Reviewer #1: A minor concern: the authors should stress that the PK properties of the drug are atypical for benzimidazole anthelmintics and enable much higher systemic exposure than attained with albendazole, for example, which is included in combination therapy for LF and has been investigated for other systemic helminth infections, including as a way to reduce microfilaremia in Loa infections. Thus, one would anticipate higher efficacy with oxfendazole, assuming intrinsic potency is similar to other BZs.

Reviewer #2: I would encourage the authors to extend the background and discussion about animal health as well. To prevent canine heartworm disease, the migrating larvae need to be treated. This could also lead to an outlook on what the treatment against L4 larvae would look like. Furthermore, the authors may want to discuss the dose-response comparing the larvicidal versus the adulticidal effects to a certain degree (particularly as parasite isolate, mouse strain and drug are uniform).

Minor remark: The graphs and abstract do not clearly state that the treatment is 2 times per day.

**Summary and General Comments**

Reviewer #1: The study is properly and carefully done. Really, my only concern is its relevance for therapy.

Reviewer #2: This is a well‑executed and valuable study that provides the first in vivo evidence of Oxfendazole’s efficacy against infective larvae of Litomosoides sigmodontis. The results are clear, the methodology is appropriate, and the findings significantly advance the understanding of Oxfendazoles therapeutic spectrum potential.

With minor revisions to strengthen context and clarity, the manuscript is well suited for publication.

PLOS authors have the option to publish the peer review history of their article (what does this mean?). If published, this will include your full peer review and any attached files.). If published, this will include your full peer review and any attached files.). If published, this will include your full peer review and any attached files.). If published, this will include your full peer review and any attached files.

...

Reviewer #1: No

Reviewer #2: **Yes:** Daniel KulkeDaniel KulkeDaniel KulkeDaniel Kulke

**Figure resubmission:** While revising your submission, we strongly recommend that you use PLOS’s NAAS tool (https://ngplosjournals.pagemajik.ai/artanalysis) to test your figure files. NAAS can convert your figure files to the TIFF file type and meet basic requirements (such as print size, resolution), or provide you with a report on issues that do not meet our requirements and that NAAS cannot fix.

**Reproducibility:**To enhance the reproducibility of your results, we recommend that authors of applicable studies deposit laboratory protocols in protocols.io, where a protocol can be assigned its own identifier (DOI) such that it can be cited independently in the future. Additionally, PLOS ONE offers an option to publish peer-reviewed clinical study protocols. Read more information on sharing protocols at https://plos.org/protocols?utm_medium=editorial-email&utm_source=authorletters&utm_campaign=protocolsTo enhance the reproducibility of your results, we recommend that authors of applicable studies deposit laboratory protocols in protocols.io, where a protocol can be assigned its own identifier (DOI) such that it can be cited independently in the future. Additionally, PLOS ONE offers an option to publish peer-reviewed clinical study protocols. Read more information on sharing protocols at https://plos.org/protocols?utm_medium=editorial-email&utm_source=authorletters&utm_campaign=protocols

---

## [Editor Report · Decision Letter 1]

13 Apr 2026

Dear Dr Hübner,

We are pleased to inform you that your manuscript 'Efficacy of oxfendazole treatment against infective larvae of *Litomosoides sigmodontis*' has been provisionally accepted for publication in PLOS Neglected Tropical Diseases.' has been provisionally accepted for publication in PLOS Neglected Tropical Diseases.' has been provisionally accepted for publication in PLOS Neglected Tropical Diseases.' has been provisionally accepted for publication in PLOS Neglected Tropical Diseases.

Best regards,

Bruce A. Rosa

Academic Editor

Lucienne Tritten

Section Editor

Shaden Kamhawi

co-Editor-in-Chief

Paul Brindley

co-Editor-in-Chief

The authors have sufficiently addressed reviewer concerns and improved the manuscript accordingly.

---

## [Editor Report · Acceptance letter]

Dear Professor Hübner,

We are delighted to inform you that your manuscript, "Efficacy of oxfendazole treatment against infective larvae of Litomosoides sigmodontis," has been formally accepted for publication in PLOS Neglected Tropical Diseases.

Best regards,

Shaden Kamhawi

co-Editor-in-Chief

Paul Brindley

co-Editor-in-Chief
